# Forecasting Molecular Features in IDH-Wildtype Gliomas: The State of the Art of Radiomics Applied to Neurosurgery

**DOI:** 10.3390/cancers15030940

**Published:** 2023-02-02

**Authors:** Rosa Maria Gerardi, Roberto Cannella, Lapo Bonosi, Federica Vernuccio, Gianluca Ferini, Anna Viola, Valentina Zagardo, Felice Buscemi, Roberta Costanzo, Massimiliano Porzio, Evier Andrea Giovannini, Federica Paolini, Lara Brunasso, Giuseppe Roberto Giammalva, Giuseppe Emmanuele Umana, Antonino Scarpitta, Domenico Gerardo Iacopino, Rosario Maugeri

**Affiliations:** 1Neurosurgical Clinic, AOUP “Paolo Giaccone”, Post Graduate Residency Program in Neurologic Surgery, Department of Biomedicine Neurosciences and Advanced Diagnostics, School of Medicine, University of Palermo, 90127 Palermo, Italy; 2Section of Radiology, Department of Biomedicine, Neuroscience and Advanced Diagnostics (BiND), University of Palermo, 90127 Palermo, Italy; 3Department of Health Promotion, Mother and Child Care, Internal Medicine and Medical Specialties, (PROMISE), University of Palermo, 90127 Palermo, Italy; 4Institute of Radiology, DIMED, University Hospital of Padova, 35121 Padova, Italy; 5Department of Radiation Oncology, REM Radioterapia srl, 95029 Vaigrande, Italy; 6Neurology Unit, ASP Trapani, 91026 Mazara del Vallo, Italy

**Keywords:** glioblastoma, brain tumor, WHO classification, TERTp, EGFR, IDH, aneuploidies, radiomics

## Abstract

**Simple Summary:**

The prognostic expectancies of patients affected by glioblastoma have remained almost unchanged during the last thirty years. Along with specific oncological research and surgical technical alternatives, corollary disciplines are requested to provide their contributions to improve patient management and outcomes. Technological improvements in radiology have led to the development of radiomics, a new discipline able to detect tumoral phenotypical features through the extraction and analysis of a large amount of data. Intuitively, the early foreseeing of glioma features may constitute a tremendous contribution to the management of patients. The present manuscript analyzes the pertinent literature regarding the current role of radiomics and its potentialities.

**Abstract:**

Background: The fifth edition of the WHO Classification of Tumors of the Central Nervous System (CNS), published in 2021, marks a step forward the future diagnostic approach to these neoplasms. Alongside this, radiomics has experienced rapid evolution over the last several years, allowing us to correlate tumor imaging heterogeneity with a wide range of tumor molecular and subcellular features. Radiomics is a translational field focused on decoding conventional imaging data to extrapolate the molecular and prognostic features of tumors such as gliomas. We herein analyze the state-of-the-art of radiomics applied to glioblastoma, with the goal to estimate its current clinical impact and potential perspectives in relation to well-rounded patient management, including the end-of-life stage. Methods: A literature review was performed on the PubMed, MEDLINE and Scopus databases using the following search items: “radiomics and glioma”, “radiomics and glioblastoma”, “radiomics and glioma and IDH”, “radiomics and glioma and TERT promoter”, “radiomics and glioma and EGFR”, “radiomics and glioma and chromosome”. Results: A total of 719 articles were screened. Further quantitative and qualitative analysis allowed us to finally include 11 papers. This analysis shows that radiomics is rapidly evolving towards a reliable tool. Conclusions: Further studies are necessary to adjust radiomics’ potential to the newest molecular requirements pointed out by the 2021 WHO classification of CNS tumors. At a glance, its application in the clinical routine could be beneficial to achieve a timely diagnosis, especially for those patients not eligible for surgery and/or adjuvant therapies but still deserving palliative and supportive care.

## 1. Introduction

Gliomas are the most common and life-threatening primary malignant brain tumors. In their most aggressive variant, namely glioblastoma, the typical biological behavior leads to a poor outcome, given that the life expectancy of these patients has not improved over the last thirty years, despite multimodal approaches and extensive research efforts [1,2,3,4,5,6,7,8,9,10,11].

Over time, palliative care has gained more importance in the context of multimodal glioblastoma management [12], as a dedicated consultation may design the best path of supportive care with a beneficial impact on patients and healthcare system organization, i.e., hospice admission seems to be higher among patients not referred to palliative care [13].

Conventional diagnostic imaging such as contrast-enhanced computed tomography (CT) and magnetic resonance imaging (MRI) constitutes the most relevant imaging modality for the initial diagnosis and staging, whereas technological facilities with high magnetic fields and advanced techniques such as spectroscopy and perfusion have increased the accuracy in the characterization of these tumors. On this imaging basis, a presumptive preoperative diagnosis is commonly attempted and preoperative planning coherently follows. Recently, radiomics and computational machine learning have been applied to glioma categorization through the extraction of a large amount of image features able to detect tumor phenotypical features.

Radiomics allows the extraction of quantitative features from conventional medical images, thus capturing tissue features and lesion characteristics, such as heterogeneity and shape, that cannot be assessed by the radiologist’s eye [14]. A typical radiomics workflow consists of the following four steps: (I) image acquisition, (II) tumor segmentation, (III) quantitative imaging feature extraction and (IV) predictive model construction and validation [15]. From a technical point of view, radiomic features could be subdivided into statistical-based, including histograms, and texture-based, the first resulting from the intensity distribution analysis of pixels/voxels within a region of interest (ROI), and the latter from the analysis of consecutive pixels/voxels; they also include model-based, aimed to interpret spatial grey-level information; transform-based, to analyze grey-level patterns in different spaces, and finally shape-based features, describing the geometric properties of ROIs [14].

Imaging characteristics have been matched to molecular tumor profiles to gain information that could lead to the prediction of glioma behavior. Indeed, this promising and rapidly evolving field deserves proper consideration because of its powerful potential to influence clinical practice [16].

While diagnostic means have consistently evolved, therapeutical efforts to improve patient outcomes have been stagnant for decades, and selected patients are currently candidates for biopsy only, rather than multimodal treatment including but not limited to maximal safe resection or supratotal resection [6,17,18,19,20,21,22,23,24].

As with all neurosurgical procedures, cerebral biopsy carries its own risks and drawbacks, among which is failure of the procedure in terms of an incorrect diagnosis. These limitations are mainly due to the inhomogeneous histopathological composition of gliomas, especially high-grade lesions, whereas necrotic areas have been classically considered crucial to obtain a diagnosis of glioblastoma [25,26,27]. Nonetheless, stereotactic biopsy is recommended when surgical management is not indicated [28]. Elderly and fragile patients might be more prone to worsening after hospitalization and surgical procedures, even if minimally invasive. As a matter of fact, substandard physical performance, namely a reduction in Karnofsky Performance Status, drives oncological management toward less aggressive supportive and palliative care, making adjuvant therapies unworkable.

The 2021 WHO Classification of Tumors of the Central Nervous System (CNS) mandates a detailed molecular tumor analysis to identify a grade 4 glioma (namely, glioblastoma wildtype) [29]. Adults’ gliomas are then divided into three types: astrocytoma isocitrate dehydrogenase (IDH)-mutant, oligodendroglioma IDH-mutant and glioblastoma wildtype. Glioma grouping has been redefined to embrace the classical histology and current practical taxonomy with the future perspective of molecular diagnosis [30]. Diagnostic criteria have progressively shifted toward genetic-based evidence, where molecular markers may be sufficient to diagnose grade 4 tumors. According to the most recent 2021 WHO classification of CNS tumors, along with IDH characterization and a “typical” histopathological background constituting glioblastoma habitat, further molecular characterization over telomerase reverse transcriptase promoter (TERT) expression, epidermal growth factor receptor (EGFR) amplification and aneuploidy is required to label a grade 4 glioma as glioblastoma. IDH is considered the most relevant gene that can aid in glioma prognostication, linked to poor outcomes in its native isoform (wildtype). In accordance, we will refer to glioblastoma wildtype as “glioblastoma”. IDH-mutant astrocytomas are now classified under the same group, borrowing their grading from classical non-CNS tumor grading based on Arabic numbers from 2 to 4. On the other hand, IDH-wildtype tumors require many molecular parameters to be evaluated, besides the histological ones, such as TERT or EGFR, which are considered sufficient to bestow the highest grade [10,11].

In this context, the latest WHO classification of brain tumors, and particularly of gliomas, lays the groundwork for more accurate stratification that also considers molecular and genomic features related to glioblastomas. The ability to exploit the analysis of radiomic features extrapolated from classical imaging techniques, and the development of AI or deep learning algorithms, aim to predict preoperatively and noninvasively molecular information critical for proper patient stratification and prognostic framing. All this is performed from the perspective of ultra-personalized medicine, intended to achieve the optimal management of each individual patient and pathology [31].

Clearly, this new classification is expected to strongly impact how the diagnostic path is conducted from now on.

Given the change in pace consequent to this new grouping of CNS tumors, it is relevant to analyze how current knowledge in radiomics may contribute to glioblastoma diagnosis and the identification of molecular features [32].

This review will provide an analysis of the state-of-the-art of literature regarding radiomics applied to glioblastoma, with the aim to understand its usefulness for the assessment of molecular features, discussing its implications for neurosurgical management.

## 2. Materials and Methods

### 2.1. Search of the Literature

The Preferred Reporting Items for Systematic Reviews and Meta-Analyses guidelines (PRISMA) were followed to conduct and report this review (Figure 1). A literature review was performed on the PubMed, MEDLINE and Scopus databases using the following keywords: “radiomics and glioma”, “radiomics and glioblastoma”, “radiomics and glioma and IDH”, “radiomics and glioma and TERT promoter”, “radiomics and glioma and EGFR”, “radiomics and glioma and chromosome”. We searched for studies published up to 17 March 2022 without backward limits. To avoid the potential omission of relevant studies, we also manually screened the reference lists of articles included and previous systematic reviews and meta-analyses regarding the same topic. Duplicate records were excluded.

### 2.2. Study Selection

The research strategy initially relied on title and abstract analysis. The article’s full text was retrieved for further investigation if the title and abstract met the inclusion criteria. The data collection process was conducted without using any automated tools. No ethical approval was required for this study.

### 2.3. Eligibility Criteria

We limited the search to those papers focusing on wildtype glioblastoma or where this entity was analyzed according to our goal and clearly distinguished from other gliomas.

Meta-analyses, reviews and clinical series were included. Non-English works were excluded. After the initial identification, each article was skimmed according to its adherence to the goal of this paper, namely the ability of current radiomics technology to detect molecular features implied in glioblastoma diagnosis. Where an article included the examination of several histotypes, thus constituting a potential source of heterogeneity, we preliminarily chose to consider only papers where wildtype glioblastoma was independently analyzed.

### 2.4. Data Extraction

Five independent reviewers were involved in the selection process. In case of discrepancies among reviewers’ findings during the literature search and analysis, a shared assessment of each paper was obtained under the supervision of the first author. The extracted data included the following: author, publication year, patient number, study design, mean age of patients enrolled, molecular findings, imaging techniques used, types of sequences used and sensibility and specificity degree obtained.

## 3. Results

### 3.1. Data Selection

A total of 719 articles were considered pertinent from our literature search and carefully evaluated to assess their eligibility. After a first analysis, 258 articles were excluded due to duplicated records. Out of 461 records, 167 were found pertinent to our topic and underwent qualitative analysis. At this point, a detailed analysis of the remaining manuscripts led us to exclude those dealing with glioma other than glioblastoma, as well as those articles not adherent to the target of our analysis. Articles not retrievable in the English language were excluded. Finally, we were able to include 11 papers in our analysis evaluating the role of radiomics for predicting molecular features in glioblastomas (Table 1).

### 3.2. Patients’ Demographic Data and Study Characteristics

We analyzed a total number of 2034 patients. All studies included in our review were retrospective studies. One study evaluated radiomic features extracted from dynamic [18F] FET PET images for the prediction of TERTp-mutation status in patients with IDH-wildtype high-grade glioma, pointing out how radiomics based on features extracted from dynamic [18F]FET PET can predict the TERTp-mutation status of IDH-wildtype diffuse astrocytic high-grade gliomas with high accuracy preoperatively. Two other studies focused on predicting TERT promoter mutation in wildtype HGG patients, this time relying on the radiomic analysis of features extracted from various MRI sequences. Interestingly, Tian and colleagues also used data extracted from MR spectroscopy, showing that the use of metabolic data increases the sensitivity and specificity of TERTp mutation prediction [35]. In fact, it is clear from their results that the radiomic nomogram based on multiparametric MR has higher prediction accuracy. Instead, seven of the studies included in our review focused on the identification of EGFR mutations through the radiomic analysis of various MRI sequences. For instance, Rathore et al. [37] systematically investigated imaging heterogeneity in patients with de novo glioblastoma, by radiomic analysis of preoperative multiparametric MRI data, hypothesizing that pattern analysis methods applied to mpMRI would be able to identify complex and otherwise visually difficult to appreciate imaging subtypes of glioblastoma that relate to the prognosis and underlying molecular characteristics of the tumor. Intriguingly, they identified three distinct subtypes showing differential characteristics in terms of overall survival rates, anatomical location, molecular composition and radiological measures of cell density, vascularization, infiltration and the extent of the tumor. They suggested that radiomen analysis can provide a more precise diagnosis, as well as more accurate prognostication. Moreover, from a personalized treatment perspective, their results indicate that subtype-specific treatments might be more effective than current standard-of-care approaches. Akbari et al. also, in their study, integrated diverse imaging features, including the tumor’s spatial distribution pattern, via support vector machines, to construct an imaging signature of EGFRvIII, which represents a driver mutation and potential therapeutic target in glioblastoma [38]. From their radiomics analysis, an imaging signature of EGFRvIII was found, revealing a distinct macroscopic glioblastoma phenotype. They pointed out the importance of this signature because it could preoperatively stratify patients for EGFRvIII-targeted therapies, and potentially monitor dynamic mutational changes during treatment. Finally, only one study focused on the ability to predict possible genetic biomarkers associated with glioblastoma, specifically 7/10 aneuploidy, using the radiomic analysis of MRI images and AI algorithms. Specifically, Calabrese and colleagues examined nine molecular biomarkers, including some that are known to affect prognosis and clinical management [43]. They found that automatically extracted radiomics features were highly sensitive for detecting aneuploidies of chromosomes 7 and 10. These aneuploidies are among the most frequent genetic alterations in glioblastoma and have been associated with malignant cell proliferation, tumor progression and lower overall survival.

## 4. Discussion

The fifth edition of the WHO 2021 Nosological Organization of CNS Tumors represents the current evolution of the field toward future more accurate classification, given that tumor diagnosis is shifting from histology to immunochemistry and molecular biology and DNA analysis [44,45,46,47,48]. Namely, the resulting hybrid taxonomy groups tumors according to mixed criteria, such as their common genetic features, i.e., IDH status.

Of note, the 2016 WHO classification identifies IDH-mutant diffuse astrocytic tumors as different entities (diffuse astrocytoma, anaplastic astrocytoma and glioblastoma), mainly according to histological parameters [49]. Recently, the 2021 WHO classification grouped all IDH-mutant diffuse astrocytomas under a single type, graded from 2 to 4, similar to the tumor grading classically adopted for non-CNS tumors. Moreover, molecular features may determine a grade 4 attribution independently of histological features such as necrosis [29].

Accordingly, a glioblastoma diagnosis is supposed to require, obviously, a setting of IDH-wildtype diffuse and astrocytic glioma, along with microvascular proliferation or necrosis, TERT promoter mutation, EGFR gene amplification or a gain of chromosome 7 and loss of chromosome 10.

The current diagnostic workflow requires a histopathological examination in any case, with the rare exception of selected patients unable to tolerate a biopsy [28]. Consequently, they may usually be considered for palliative care, since these patients often do not meet the criteria of eligibility for radio-chemotherapy. Early palliative management may improve patients’ quality of life and should be provided as soon as possible where other therapeutic strategies could not be recommended [50]. Patients adequately followed up could benefit from the early initiation of supportive therapies, once they are no longer eligible for other treatments. The correct management allows patients to be admitted on time to every step of the pathway, transitioning from surgery to radio-chemotherapy to palliative care, providing the best standard of care currently available [13].

Depending on the tumor size, location and presumptive diagnosis on the basis of conventional radiological imaging, taking into account also the patient’s condition in terms of performance status, a surgical resection is considered part of the gold-standard management [7,8,21,51,52,53].

When the tumor features and/or patient condition discourage surgery, a cerebral biopsy is generally performed. Glioblastoma is well known not only for its infiltrative, aggressive spreading through parenchymal tissue, but also for being a highly heterogeneous and rapidly evolving lesion in terms of its size and pace of growth, determining wide necrotic areas in this context. Histopathological examination on a tumor sample coming from a biopsy may then not be diagnostic or may underestimate the tumor grade since it represents only a small sample of the lesion [42,54]. Moreover, as with every surgical procedure, a cerebral biopsy requires general anesthesia and carries specific risks linked to the procedure itself.

Ideally, we suggest that since a cerebral biopsy often helps in reaching a diagnosis, generally determining poor overall survival, further efforts should be encouraged to achieve the same result in the least invasive way possible. In fact, glioblastoma has poor overall survival despite multimodal management.

Recently, conventional diagnostic techniques have been placed alongside an emerging translational field where an array of different features are analyzed along with image textures to decode the phenotypic patterns of tumors [55,56]. Radiomics is a translational field focused on decoding conventional imaging data to extrapolate the molecular features of tumors such as gliomas [16].

Radiomics may improve clinical decision-making as it can differentiate between tumor grades, identify druggable mutations and eventually assess the tumor response [57,58]. It may help to promptly identify patients with a poor diagnosis and potentially impact their next treatment decision [59,60]. Considering the wide heterogeneity of gliomas, one of the major advantages of radiomics resides in the evaluation of the entire tumor volume, avoiding potential errors coming from the analysis of a small tumor sample obtained by tumor biopsy. In this scenario, radiomics could also help in providing a noninvasive diagnosis for selected patients, whereas invasive diagnostic strategies appear not reasonable due to patients’ global clinical status.

Although discussing IDH detection through radiomics is beyond the scope of our work, it is worth mentioning that radiogenomics has represented a useful tool to forecast and classify patients with low-grade and high-grade gliomas according to risk groups, thanks to the evaluation of IDH mutation, along with 1p19q codeletion. Several reports have highlighted the importance of “T2-FLAIR mismatch”, i.e., a hyperintensity on T2-weighted imaging and a hypointense signal on FLAIR sequences, as a means to identify IDH-mutant astrocytoma with 100% specificity [1,2].

To date, a more accurate prognostication has been researched and obtained using a combined analysis of IDH mutation and imaging. Three different tumor subtypes, related to IDH1-wildtype gliomas, have been identified: the “solid” one, characterized by small peritumoral infiltration, that can benefit from radical tumor resection; the “irregular” one, characterized by invasive behavior and a poor response to aggressive treatments, often located in the temporal lobe; and the “rim-enhancing” subtype, less vascularized, with a lower cell density and an enhanced peripheric rim. It is worth noting that overall survival has been found similar in these “rim-enhancing” subtypes when comparing IDH wildtype and IDH mutant tumors, which may guide their treatment [9].

TERTp may undergo mutations, leading to lengthened telomeres, whose role in aggressive brain cancer development has been recognized, obviously along with a clear association with poor overall survival [30,33,61,62,63,64,65,66,67]. For these reasons, TERT profiling has been recently included among the pivotal glioblastoma molecular features [29,68,69] and its noninvasive preoperative identification has been attempted. Most of the radiomics studies focusing on MRI have analyzed WHO grade II and/or WHO grade III gliomas (grouped following previous WHO classification), reporting encouraging rates of accuracy (up to 93.8%) [34,35,70,71,72,73]. In a retrospective study, Tian et al. [57] constructed a radiomics score for the prediction of TERT promoter mutations in 126 patients with high-grade glioma, demonstrating excellent performance in both training and validation cohorts. Yan and coworkers [56] incorporated radiomics features from contrast-enhanced T1-weighted imaging and apparent diffusion coefficients to predict IDH and TERT mutational status. In their study, the resulting radiomics model allowed them to construct a predictive nomogram with good performance for the stratification of patients according to progression-free survival and overall survival [56].

Furthermore, Li et al. extracted radiomics features from dynamic [18F]FET PET to predict TERTp-mutation status in 159 patients with glioblastoma [63].

EGFR aberrant signaling has been a major topic in oncology for decades. In particular, its gene amplification and overexpression can be observed in almost 40% of glioblastomas, most frequently in primary forms (“de novo”). These tumors show marked angiogenesis and tend to invasively infiltrate the surrounding brain parenchyma, being partially resistant to radiotherapy and more prone to recurrence after multimodal treatment [74,75,76,77]. In almost half of these cases, a specific EGFR mutant, the so-called EGFR variant III (EGFRvIII), can be recognized. EGFRvIII generates a specific downstream signal pathway that strongly affects tumorigenicity [78]. In addition, EGFR amplification (especially EGFRvIII) and TERT mutation have gained a key role in recognizing low-grade glioma IDH-wildtype, potentially associated with the worst prognosis and more aggressive behavior [3,4,5]. EGFR amplification is commonly related to relative cerebral blood volume (rCBV) textural features, particularly with microvessel volume and angiogenesis [6]. Therefore, the use of radiomics can aid in detecting these mutations before surgery, even if it has been demonstrated that TERT promoter mutations are tumor-related, and forecast patients’ prognosis, regardless of IDH status, despite showing, in some studies, lower accuracy compared to IDH and 1p/19q codeletion [7,8]. Kihira et al. [66] provided an MRI-based radiomics model with an AUC of 0.83 for the identification of EGFR mutant gliomas by combining three main texture features. Rathore et al. [67] used a combined multiparametric approach to identify EGFRvIII-mutated glioblastomas in a study involving 261 patients. Akbari and colleagues [68] integrated multiple imaging features into a support vector machine model, which achieved sensitivity of 78.6% and specificity of 90% for the diagnosis of EGFRvIII mutations. Similarly, Pasquini et al. [69] obtained accuracy of 81% for the detection of EGFR amplification using a machine learning model. In a large study including 418 patients with pathologically proven glioblastomas, Sohn et al.’s radiomics model demonstrated sensitivity of 81.2%, specificity of 58.5% and an AUC of 0.743 for EGFR amplification prediction [70].

Likewise, chromosome aberrations have been the focus of radiomics applications, albeit mostly regarding chromosome 1p/19q co-deletion rather than aneuploidies of chromosome 7 and 10, whose status has gained relevance in accordance with the recently released WHO classification of CNS tumors.

Aneuploidies of chromosomes 7 and 10 are frequently detected in glioblastoma, often conditioning its aggressiveness due to rapid tumor progression and poor overall survival [79,80]. Despite relatively little existing work, radiomics results regarding the detection of aneuploidies seem promising [43] Calabrese et al. [65] provided a fully automated artificial intelligence method with sensitivity of 94% and specificity of 88% when predicting aneuploidies in 199 patients with glioblastoma. Nevertheless, the exact pathway leading to genetic aberrations in glioblastoma, such as aneuploidy, is scarcely known and its understanding appears of paramount importance to clarify the tumorigenicity process.

### Limitations of the Study

The present study considered a wide number of papers regarding glioblastoma biological phenotypes. This is a highly prolific field in the literature, with rapid evolution in the number of papers retrievable and most of all in the content of scientific knowledge. Our effort was intended to capture the current literature findings at the present time, being aware that today’s state-of-the-art might be outdated within a short time.

The 2021 WHO classification of CNS tumors has focused on precise criteria identifying glioblastoma, whereas molecular features gain the utmost importance, thus indirectly encouraging radiomic studies. Nevertheless, all the above-mentioned studies referred to the 2016 WHO classification considering gliomas according to their previous grading system. The latest 2021 classification undoubtedly requires radiomics studies to be updated in accordance.

## 5. Conclusions

Glioblastoma remains one of the most aggressive brain tumors, burdened by poor outcomes despite multimodal therapy. Research efforts have classically focused on stratifying patients according to clinical parameters and phenotypical features, with the aim to disclose crucial prognostic information. The most recent WHO classification of CNS brain tumors recommends glioblastoma diagnosis to be carried out with molecular targets, along with well-known histopathological features.

Radiomics is a translational field focused on decoding conventional imaging data to extrapolate the molecular features of tumors such as gliomas. Its application to glioblastoma may improve our knowledge and lead toward the tailored management of each single case. In particular, radiomics could ease patient selection for targeted therapies and allow the continuous monitoring of possible molecular changes during treatment, as well as help in quantifying molecular modifiers’ fluctuations after therapies. Likewise, it could represent a valuable tool to obtain a diagnosis, bypassing interventional strategies in fragile, well-selected patients, whereas palliative and supportive care, when promptly started, may significantly impact their quality of life and, finally, facilitate the decent management of quality of life issues.

The analysis of current radiomics’ ability to uncover molecular targets required for a glioblastoma diagnosis shows that, although much has been achieved, further efforts are required to focus on the aforementioned new molecular targets. Accordingly, shared radiomics operative models are desirable to make tumor analysis somewhat homogeneous among many centers and thus progress towards radiomics as a reliable technique in everyday practice and clinical workflows.

## Figures and Tables

**Figure 1 cancers-15-00940-f001:**
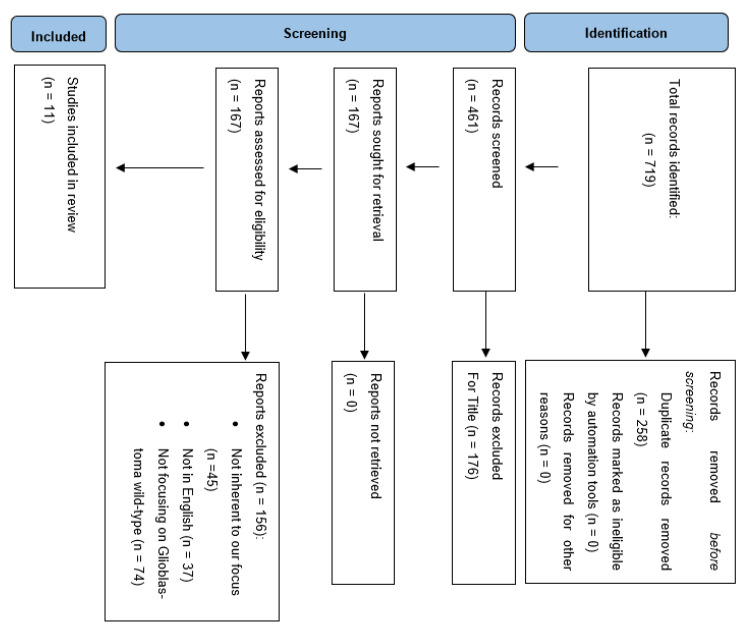
PRISMA flow chart of selection process.

**Table 1 cancers-15-00940-t001:** Synthesis and comparison of the papers selected and included in review after application of the PRISMA flow chart.

Author/Year	Study Design	Patients Enrolled	Mean Age	Molecular Finding (TERT, EGFR, Aneuploidy)	Imaging Techniques	Sequences	Best Sens/Spec/AUC Reached
Z. Li et al., 2021 [33]	Randomized controlled trial	159	60.2	TERTp mutations	PET	Dynamic [18F]FET PET	0.921/NA/0.82
J. Yan et al., 2021 [34]	Retrospective study	357	N/A	TERTp mutations	MRI	CE-T1w, DWI (using ADC)	0.944/0.400/0.811
H. Tian et al., 2020 [35]	Retrospective study	126	N/A	TERTp mutations	MRI	CE-T1w, T1w, T2w, T2-FLAIR, MRS	0.947/0.840/0.955
S. Kihira et al., 2021 [36]	Retrospective study	111	57.0	EGFR amplification	MRI	CE-T1W, T2–FLAIR, DWI.	0.65/0.68/0.83
S. Rathore et al., 2018 [37]	Retrospective study	208	N/A	EGFRvIII	MRI	CE-T1w, T1w, T2w, T2-FLAIR, DSC MRI	NA
H. Akbari et al., 2018 [38]	Retrospective study	129	59.3	EGFRvIII	MRI	CE-T1w, T1w, T2w, T2-FLAIR, DTI, DSC MRI	0.786/0.90/0.86
Pasquini L. et al., 2021 [39]	Retrospective study	156	N/A	EGFR amplification	MRI	MPRAGE, T1w, T2w, T2-FLAIR, DWI, DSC MRI	NOTE: accuracy 81%; ROC 74.3%.
B. Sohn et al., 2021 [40]	Retrospective study	418	60.1	EGFR amplification	MRI	CE-T1w, T1w, T2w, T2-FLAIR	0.812/0.585/0.743
O. Zinn et al., 2017 [41]	Retrospective study	29	N/A	EGFR	MRI	CE-T1w, T1w, T2w, T2-FLAIR	NA
S. Bakas et al., 2017 [42]	Retrospective study	142	59.82	EGFRvIII	MRI	CE-T1w, T2-FLAIR, DSC MRI	0.8377/0.9235/0.8869
Calabrese E. et al., 2020 [43]	Retrospective study	199	N/A	Aneuploidy	MRI	T2w, T2-FLAIR, SWI, DWI, CE-T1w, T1w, ASL perfusion images, HARDI	0.90/0.88/0.93

## Data Availability

Pubmed (https://pubmed.ncbi.nlm.nih.gov/), Medline (https://www.nlm.nih.gov/medline/index.html), Scopus (https://www.scopus.com/home.uri).

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
