# Peer review of "Forecasting Molecular Features in IDH-Wildtype Gliomas: The State of the Art of Radiomics Applied to Neurosurgery"

_cancers, 2023, doi:10.3390/cancers15030940_

Round 1

Reviewer 1 Report

Review

In their manuscript „Forecasting molecular features in IDH‑wildtype gliomas: the state of the art of radiomics applied to neurosurgery.” the authors screened the literature for articles describing radiomics as a tool to predict molecular markers in glioblastomas.

Comments:

Introduction:

In the introduction the authors explain todays diagnostic and therapeutic pathway and the importance of new molecular markers included in the WHO’s Classification of Tumors as well as a short introduction on how radiomics works.

Line 55: double space

Line 73: “tht”

Line 84: “field” instead of “filed”

Methods:

The authors performed literature research according to a defined set of keywords on 3 different platforms. It should be also mentioned in which timespan or to which date articles were included as radiomics is a fairly fast evolving field of ongoing research.

It would be helpful to provide the reader with a flowchart according to the PRISMA guidelines as the authors describe in the headline of table 1. It should inform the reader about the different stages of the selection process.

Results:

In the results section the selection process is briefly described and a table with the 11 articles finally selected is given together with a short summary.

In the table provided by the authors they state having applied a PRISMA flowchart. The acronym PRISMA (TRANSPARENT REPORTING of SYSTEMATIC REVIEWS and META-ANALYSES ) and the workflow should be briefly explained. There is no flowchart in the document describing on how the review process was organized.

In the results section the different articles are not presented to the reader in detail. Instead they are briefly explained in the Discussion section. I suggest to explain the different articles more in detail and sort them by topic with different paragraphs (EGFR, TERT, Aneuploidy) as well as move this from the discussion to the results sections. In comparison to the length of the whole manuscript the explanation of the authors’ “findings” namely the selected articles after the PRIMSMA selection is fairly short (line 237-291) and should be explained and discussed in a more detailed way while other reduntant parts might be shortend.

Discussion:

In the beginning of this section the authors mainly repeat the importance of molecular markers in the diagnosis of gliobastoma from the introduction. I would propose to shorten this paragraph.

Furthermore the citations of the articles of table 1 are not the same order as in the text (e.g. line 252 “Li et al.” [62] vs “Li et al.”[32] in the table).

Conclusion

I would recommend this article for publication after revision. It should be structured more clearly especially in the results and discussion section as mentioned above.

Reviewer 2 Report

the authors should not focus on the 2021 revision of the WHO, since their review does not include papers relating to this revision

I recommend to analyze how the new WHO classification might impact the review

Round 2

Reviewer 2 Report

-